# Metabolic Engineering for Efficient Production of Z,Z-Farnesol in *E. coli*

**DOI:** 10.3390/microorganisms11061583

**Published:** 2023-06-15

**Authors:** Mengyang Lei, Zetian Qiu, Leilei Guan, Zheng Xiang, Guang-Rong Zhao

**Affiliations:** 1Frontiers Science Center for Synthetic Biology, Key Laboratory of Systems Bioengineering (Ministry of Education), School of Chemical Engineering and Technology, Tianjin University, Yaguan Road 135, Jinnan District, Tianjin 300350, China; mylei@tju.edu.cn (M.L.); ztqiu@tju.edu.cn (Z.Q.); guan@tju.edu.cn (L.G.); 2Georgia Tech Shenzhen Institute, Tianjin University, Dashi Road 1, Nanshan District, Shenzhen 518055, China; 3State Key Laboratory of Chemical Oncogenomics, School of Chemical Biology and Biotechnology, Peking University Shenzhen Graduate School, Shenzhen 518055, China

**Keywords:** (2Z,6Z)-farnesol, enzyme engineering, metabolic engineering, *Escherichia coli*, synthetic biology

## Abstract

Z,Z-farnesol (Z,Z-FOH), the all-*cis* isomer of farnesol, holds enormous potential for application in cosmetics, daily chemicals, and pharmaceuticals. In this study, we aimed to metabolically engineer *Escherichia coli* to produce Z,Z-FOH. First, we tested five Z,Z-farnesyl diphosphate (Z,Z-FPP) synthases that catalyze neryl diphosphate to form Z,Z-FPP in *E. coli*. Furthermore, we screened thirteen phosphatases that could facilitate the dephosphorylation of Z,Z-FPP to produce Z,Z-FOH. Finally, through site-directed mutagenesis of *cis*-prenyltransferase, the optimal mutant strain was able to produce 572.13 mg/L Z,Z-FOH by batch fermentation in a shake flask. This achievement represents the highest reported titer of Z,Z-FOH in microbes to date. Notably, this is the first report on the de novo biosynthesis of Z,Z-FOH in *E. coli*. This work represents a promising step toward developing synthetic *E. coli* cell factories for the de novo biosynthesis of Z,Z-FOH and other *cis*-configuration terpenoids.

## 1. Introduction

Terpenoids are an immensely diverse and abundant class of natural compounds composed of multiple isoprene units [1]. Examples of well-known terpenoids include limonene, artemisinin, paclitaxel and carotenoid. These compounds exhibit a wide range of physical and physiological properties, such as pleasant fragrances, high energy density and heating value, and antibacterial, anti-inflammatory, and analgesic effects [2,3,4]. As a result, they have extensive applications in various industries, including food, cosmetics, agriculture, pharmaceuticals, and energy [5]. However, the limited amount of terpenoids produced in plants makes extraction and purification inefficient and hence expensive. Additionally, the chemical synthesis of terpenoids is environmentally unfriendly, along with the formation of impure products [6]. Therefore, synthetic biology provides a promising alternative for the efficient and sustainable production of terpenoids through microbial cell factories [7].

*Escherichia coli* is a suitable host for terpenoid production due to its rapid growth and ease of genetic manipulation. Researchers have already achieved gram-scale production of limonene [8], farnesene [9] and amorphadiene [10] in *E. coli* using fermentable carbon sources. The biosynthesis of terpenoids begins with the universal building blocks, dimethylallyl diphosphate (DMAPP) and isopentenyl diphosphate (IPP), which are generated by the mevalonate (MVA) and methylerythritol phosphate (MEP) pathways [11,12]. These building blocks are condensed to form long isoprenyl diphosphates, such as geranyl diphosphate (GPP) and farnesyl diphosphate (FPP), which serve as precursors for the synthesis of monoterpenes and sesquiterpenes [13].

In nature, there exist three main stereoisomers of FPP, namely E,E-FPP, Z,E-FPP, and Z,Z-FPP (Appendix A) [5]. The conformational geometry of their two double bonds (C2=C3 and C6=C7) determines the stereochemistry of these isomers. All E-double bond-forming enzymes such as IspA [14] in *E. coli* and Erg20 [15] in *S. cerevisiae* catalyze the formation of E,E-FPP, which serves as a precursor and substrate for the synthesis of common trans-terpenoids such as artemisinin and squalene [16]. In contrast, Z,E-FPP has more advantages in the cyclization of some terpenoids, such as *endo*-bergamotene, due to stereochemical constraints in the reaction [17]. Z,E-FOH has already been produced in *E. coli* using a fusion enzyme of IspA and Z,E-FPP synthase (Rv1086) [14]. Z,Z-FPP is synthesized by *cis*-prenyltransferases (CPTs) and is involved in the biosynthesis of long-chain terpenoids, ranging in size from C10 to C1000 [18,19]. A well-known example of such a terpenoid is natural rubber, which is composed mainly of cis-1,4-polyisoprene [20].

Advances in synthetic metabolism research have led to the identification and analysis of Z-type terpene synthesis pathways and their encoded enzymes in microorganisms, insects, and plants. PsIDS3 [21] is a multifunctional enzyme, that was identified in the crucifer flea beetle *Phyllotreta striolata*. PsIDS3 was found to produce neryl diphosphate (NPP) when supplied with DMAPP and IPP, and to utilize NPP and IPP to synthesize Z,Z-FPP. When supplied with GPP and IPP, PsIDS3 produced Z,E-FPP instead. SlCPT6 [18] from tomato (*Solanum lycopersicum*) preferred NPP and IPP as substrates to produce Z,Z-FPP, while ShzFPS [22] from wild tomato (*Solanum habrochaites*) was capable of generating NPP and Z,Z-FPP when supplied with DMAPP and IPP. Notably, no product was detected when GPP was used instead of DMAPP. Rv1086 [23] from microorganism *Mycobacterium tuberculosis* and Mvan4662 [24] from *M. vanbaalenii* are reported to accept only GPP to synthesize Z,E-FPP. Additionally, monoterpenes such as limonene, terpinene, and carene in tomato glandular trichomes are synthesized from NPP rather than GPP [25]. SBS, the santalene and bergamotene synthase [22] from wild tomato *S. habrochaites* uses Z,Z-FPP as its sole substrate to generate a mixture of cyclic terpenoids [22]. Although enzymes that exhibit substrate specificity for Z,Z-FPP are still being studied, these findings provide insight into the complex and diverse nature of terpene synthesis pathways.

In this study, we aimed to construct a de novo biosynthesis pathway for Z,Z-FOH and increase its production in *E. coli* using rational engineering strategies. Initially, we screened five Z,Z-FPP synthases provided with NPP substrate in the prene-overproducing strain EIP81 (Figure 1) to produce Z,Z-FOH. We also overexpressed different phosphatases to assist in the dephosphorylation of Z,Z-FPP. Additionally, we enhanced the biosynthesis of Z,Z-FOH by protein engineering of Mvan4662 based on its structural information. Ultimately, the engineered strain ELMY101 produced the highest Z,Z-FOH titer of 572.13 mg/L in two-phase batch fermentation in a shake flask. It is the first report of Z,Z-Farnesol production in microorganisms.

## 2. Materials and Methods

### 2.1. Strains, Plasmids, and Reagents

The strains and plasmids used in this study are listed in Table 1 and Table 2, respectively. The strains and plasmids are available upon request. *Escherichia coli* DH5α was used for cloning and plasmid construction, and *E. coli* EIP81 [26] from our previous work was used as the starting strain for construction of Z,Z-FOH-producing strains. (2Z,6Z)-Farnesol standard was purchased from ZZBIO Co., Ltd. (Shanghai, China). The ClonExpress II One Step Cloning Kit, DNA Polymerase of Phanta Super Fidelity and Taq were obtained from Vazyme (Nanjing, China). Purification of DNA, gel extraction, and plasmid preparation were conducted using kits from Vazyme (Nanjing, China). The primers of the polymerase chain reaction (PCR) used in this study were synthesized by GENEWIZ (Suzhou, China) and are listed in Appendix A.

### 2.2. Construction of Plasmids and Strains

In order to better express exogenous enzymes, the membrane-targeting signal peptides at the N-terminus were deleted. Genes encoding truncated NPPS of *S. lycopersium* (SltNPPS, N-terminal 45 amino acid-truncated), Z,Z-FPPS of *M. vanbaalenii* (mvan4662), *M. tuberculosis* (rv1086), *P. striolata* (PstIDS3, N-terminal 30 amino acid-truncated), *S. lycopersicum* (SltCPT6, the N-terminal 12 amino acid-truncated), and *S. habrochaites* (ShtZFPS, the-terminal 45 amino acid-truncated) were codon-optimized for *E. coli* and synthesized by GenScript (Nanjing, China). The codon-optimized and truncated gene sequences used in this work are listed in Appendix A.

All plasmids were constructed using PCR and homologous recombination methods. Amplified *SltNPPS* fragment using PCR and linearized pACYCDuet-1 vector were assembled using the ClonExpress II One Step Cloning Kit (Vazyme, Nanjing, China) to generate plasmid pLMY28. *PstIDS3*, *mvan4662*, *rv1086*, *SltCPT6*, and *ShtzFPS* were amplified using their corresponding upstream and downstream primers and then assembled into the pRSFDuet-1 vector to generate plasmids pLMY51, pLMY52, pLMY53, pLMY61 and pLMY63, respectively.

Eleven endogenous phosphatase genes (*rdgB*, *phoA*, *ppa*, *cdh*, *lpxH*, *mutT*, *nudC*, *nudF*, *nudG*, *nudJ*, *pgpB* and *ybjG*) (Appendix A) were amplified from *E. coli* BW25113(DE3). The heterogeneous gene *ScLPP1* of *S. cerevisiae* S288C and truncated *TcNudix1* of *Tanacetum cinerariifolium* were amplified from plasmids stored in our laboratory [26]. These genes were then constructed into the other multiple cloning sites of plasmid pLMY52 to generate plasmids pLMY65 to pLMY77. Each site mutant at C41, Y85, L92, D95, E97, Q93 and L99 was introduced in the *mvan4662* gene using a standard mutation PCR procedure with two specific primers using template pLMY74. The PCR product was treated with DpnI enzyme to eliminate the influence of the template and the resulting fragments were then assembled to generate plasmids pLMY81 to pLMY113.

All plasmid constructed above were then transformed into strain ELMY28 which overproduced precursor NPP to generate Z,Z-FOH-producing strains.

### 2.3. Model Analysis of Mvan4662-Z,Z-FPP Complex by Computational Simulation

The binding model of Mvan4662 and Z,Z-FPP was analyzed by homologous modeling through the Swiss-Model program (http://swissmodel.expasy.org, accessed on 10 September 2022), with the crystal structure of the Rv1086-E,E-farnesyl diphosphate complex (PDB:2VFW) serving as the template. Additionally, molecular docking simulation of Mvan4662 and its product Z,Z-FPP were performed using Autodock vina 1.2.3 software. The results of the docking models and protein structures were visualized and analyzed using PyMOL v2.4 software.

### 2.4. Media and Cultivation Conditions

Luria-Bertani (LB) medium (10 g/L NaCl, 10 g/L tryptone and 5 g/L yeast extract) was used for plasmid propagation and seed preparation, and M9Y medium (per liter: 6.78 g/L Na_2_HPO_4_, 3 g/L KH_2_PO_4_, 1 g/L NH_4_Cl, 0.5 g/L NaCl, 5 mM MgSO_4_, 2.0g/L yeast extract and 0.1 mM CaCl_2_) supplemented with 10 g/L glucose were used for cell growth and shake-flask fermentation. As required, antibiotics were added to the culture medium at a final concentration of 100 mg/L of ampicillin, 30 mg/L of streptomycin, 30 mg/L of kanamycin and 20 mg/L of chloramphenicol. For batch fermentation in a shake flask, *E. coli* strain was first grown in 4 mL of LB liquid medium at 37 °C and 220 rpm for 12 h and then inoculated into M9Y medium containing 10 g/L glucose at 30 °C and 250 rpm for approximately 24 h. The cells were pelleted and resuspended in 50 mL of fresh M9Y medium containing 10 g/L glucose with an initial OD at 600 nm of 3 for fermentation. Meanwhile, isopropyl β-d-1-thiogalactopyranoside (IPTG) at a concentration of 0.5 mM and dodecane at a concentration of 10% (*v*/*v*) were added to the medium to induce gene expression and collect the product. The cultures were then fermented in a shake flask for 84 h at 250 rpm and 30 °C.

### 2.5. Biomass and Metabolite Analysis

Cell optical density (OD) was measured at 600 nm with a UV-5100B spectrophotometer (Metash Instruments, Shanghai, China). Residual glucose was quantified using a biosensor S-10 (Siemantec Technology, Shenzhen, China). Metabolite was quantified using a Shimadzu System GCMS-QP2020 NX equipped with a SH-Rxi-5Si1 MS capillary column (30 m × 0.25 mm and 0.25 μm film thickness). The culture was centrifuged at 10,000 rpm for 4 min and the upper organic phase was collected and dehydrated using anhydrous sodium sulfate. The water-removed sample was filtered through a 0.22 μm organic filter membrane and 1mL was collected into a sample vial. The injection volume was 1 μL without splitting and the injection temperature was maintained at 250 °C. Helium was used as the carrier gas with a line speed of 40 cm/s. The chromatographic column heating program was as follows: The initial temperature was set at 60 °C and heated to 150 °C at a rate of 15 °C/min, held for 3 min, then ramped to 200 C at a rate of 6 °C/min and held for 2 min. In the final post-processing stage, the temperature is raised to 300 °C at a rate of 20 °C/min and held for 3 min. The mass scan ranged from 35 m/z to 400 m/z. All GC analyses were quantified using a seven-point calibration curve and the calibration curve had an R^2^ coefficient higher than 0.99.

## 3. Results

### 3.1. Screening Z-FPP Synthases with NPP as the Substrate

The introduction of an exogenous MVA pathway in *E. coli* is a common strategy for obtaining sufficient IPP and DMAPP building blocks for terpenoid biosynthesis. In this study, we used the strain EIP81 as the platform for terpenoid production, this strain was previously optimized for the exogenous MVA pathway [26] (Figure 1). We used growth-orthogonal isomeric substrates NPP and Z,Z-FPP as universal precursors for Z,Z-FOH biosynthesis instead of GPP and E-FPP [27]. While NPP has been used as the specific substrate for the biosynthesis of monoterpenoids, such as limonene [2] and nerol [26], the use of Z,Z-FPP as the specific substrate for the biosynthesis of sesquiterpenoids in *E. coli* has not been reported. We overexpressed NPP synthase (*SltNPPS*) from *S. lycopersicum* in EIP81 strain, yielding strain ELMY28. Next, we searched the NCBI database to identify annotated Z,Z-FPP synthases, or cis-isoprenyl diphosphate synthases and selected candidate enzymes, including PsIDS3 from *P. striolata*, SlCPT6 from *S. lycopersicum*, ShzFPS from *S. habrochaites*, Rv1086 from *M. tuberculosis* and Mvan4662 from *M. vanbaalenii*, respectively, through PCR amplification to generate the truncated versions PstIDS3, SltCPT6 and ShtzFPS for subsequent plasmid construction. We then constructed five plasmids with these codon-optimized Z,Z-FPP synthases and introduced them into the high-NPP-producing strain ELMY28, generating strains ELMY51, ELMY52, ELMY53, ELMY61 and ELMY63. After 72 h of fermentation, gas chromatography-mass spectrometry (GC-MS) analysis revealed that all five *E. coli* strains were capable of synthesizing Z,Z-FOH (Figure 2). However, strains ELMY51 and ELMY61 exhibited the lowest Z,Z-FOH production, indicating low Z,Z-FPPS catalytic activities for PstIDS3 and SltCPT6. ELMY52, ELMY53 and ELMY63 exhibited the potential to achieve significant Z,Z-FOH production. Additionally, minor amounts of Z,E-FOH and E,E-FOH were detected, this might because endogenous GPP was used as substrate (Appendix A). After evaluating the Z,Z-FOH production of different strains, we selected ELMY52, which produced the highest Z,Z-FOH titer of 13.28 mg/L, for further investigation. Notably, Rv1086, previously identified as a Z,E-FPP synthase [14] for its ability to catalyze the conversion of GPP to Z,E-FPP, can also utilize NPP as the substrate to produce Z,Z-FPP, thereby functioning as a Z,Z-FPP synthase as well.

### 3.2. Effects of Phosphatases on Z,Z-FOH Production

Notably, the Z,Z-FOH detected in the strains under investigation was likely derived from the hydrolysis of Z,Z-FPP by some endogenous phosphatases, as previously reported for the hydrolysis of E,E-FPP to E,E-farnesol in *E. coli* [28,29]. This process can be triggered by the overproduction of FPP beyond normal levels, leading to the activation of endogenous promiscuous phosphatases and hydrolysis of FPP to FOH with the release of two phosphates [30]. To further improve the production of Z,Z-FOH, we explored the potential of phosphatases for FOH synthesis and recruited 13 hydrolases (Appendix A), including 11 endogenous hydrolases in *E. coli*: dITP/XTP pyrophosphatase (RdgB), periplasmic alkaline phosphatase (PhoA), inorganic pyrophosphatase (Ppa), CDP-diacylglycerol pyrophosphatase (Cdh), UDP-2,3-diacylglucosamine hydrolase (LpxH), 8-Oxo-dGTP diphosphatase (MutT), RNA decapping hydrolase (NduC), ADP-ribose pyrophosphatase (NudF), phosphatidylglycerophosphatase B (PgpB), undecaprenyl pyrophosphate phosphatase (YbjG) and two exogenous hydrolases, namely, the phosphatidate phosphatase LPP1 from *S. cerevisiae* (ScLPP1) and truncated chloroplast Nudix1 from *T. cinerariifolium* (TctNudix1). These hydrolase genes were introduced into strains ELMY65 to ELMY77. All the endogenous hydrolases tested increased the production of Z,Z-FOH compared to the reference strain ELMY52, which was not overexpressed phosphates (Figure 3). Among the endogenous hydrolases, YbjG and LpxH increased the production of Z,Z-FOH by 63% and 109%, respectively, while RdgB, PhoA, Ppa, Cdh, MutT, NudC and NudF enhanced the production of Z,Z-FOH by 166% to 202% more than that in ELMY52. Significantly, we achieved a substantial improvement in the Z,Z-FOH titer by overexpressing NudJ and PgpB in the ELMY74 and ELMY76 strains, respectively, resulting in increases of 709% and 514% with titers of 73.89 mg/L and 56.13 mg/L, respectively. With the exogenous hydrolases ScLPP1 and TctNudix in ELMY67 and ELMY75, Z,Z-FOH levels of 16.30 mg/L and 18.29 mg/L were achieved, representing increases of 78% and 63%, respectively, compared to the yield in strain ELMY52. This indicates that these two exogenous hydrolases did not outperform endogenous phosphatases. As a result, we selected the ELMY74 strain, which harbored SltNPPS, Mvan4662, and NudJ and produced the highest Z,Z-FOH yield, for further studies.

### 3.3. Enhancing Z,Z-FPP Production through Engineering of Mvan4662

Mvan4662 belongs to the cis-isoprenyl diphosphate synthase superfamily and is mainly responsible for linear isoprenoid production through head-to-tail condensation [31]. The homodimeric architecture of members of this family forms a butterfly like structure with the two monomers resembling butterfly wings [32], held together by hydrogen bonds and hydrophobic interactions. By engineering Mvan4662, it may be possible to enhance Z,Z-FOH production. Our goal was to optimize the performance of Mvan4662 by utilizing structural information for protein engineering. We first conducted sequence alignment (Appendix A) of six cis-isoprenyl diphosphate synthases to identify critical, strictly conserved sites for enzyme activity that should not be altered. Next, we predicted the structure of Mvan4662 through homology modeling using the crystal structure of the Rv1086-FPP complex (PDB:2VFW) as a template. Rv1086 has already been identified as a Z,E-FPP synthase [14], so we then docked the template with the products Z,Z-FPP and Z,E-FPP to identify the different and non-conserved sites. Due to the flexibility of the linear substrate structure, we predicted multiple structures for the docking complex between the protein and the ligand FPP, and we selected the three relatively stable structures (Figure 4A–C) on the basis of the binding free energy and the substrate position in the enzyme cavity. We then examined the residues within 4 Å of the carbon chain and found that the non-conserved residues C41 and Y85 differed among all three docking models (Figure 4D). Consequently, we selected C41 and Y85 as the first series of candidates for mutation. Since the amino acids around the carbon chain were mostly uncharged, we substituted these two residues with different uncharged amino acids such as Ala, Phe, Leu, Ser, and Trp.

Residues C41 and Y85 have a significant impact on the synthesis of Z,Z-FOH. When C41 was mutated to Phe, Trp, or Tyr, almost no synthesis of Z,Z-FOH occurred (Figure 4E). When C41 was mutated to Leu and Met, Z,Z-FOH production was decreased. However, when C41 was mutated to smaller volume residues such as Ala and Ser, Z,Z-FOH production increased. The C41S mutant strain showed the highest increase, up to 129.78 mg/L. When Y85 residue was mutated to smaller volume residues such as Ala, Leu, and Ser, Z,Z-FOH was significantly reduced or even lost. When Y85 was mutated to Phe, Z,Z-FOH production was increased. When it was mutated to Try, Z,Z-FOH production remained almost unchanged. Among these mutations, the Y85F mutant strain exhibited the highest Z,Z-FOH production of up to 168.56 mg/L.

Furthermore, it was reported that residues from S74 to V85 in the crystal structure of UPPs were not strictly conserved in region III, indicating that the flexible domain is important for substrate binding and catalysis [33]. Therefore, we focused on five specific sites in the flexible domain of the conserved region III of Mvan4662, namely 92L, 93Q, 95D, 97E, and 99L, which are crucial for proper IPP binding and catalytic function. We altered these five residues to match those found in the *cis*-prenyltransferases family and created a small but smart library of 27 single-point mutants to optimize Z,Z-FOH biosynthesis.

We conducted fermentation tests on the mutant strains of the second class of candidate mutation sites, namely, L92, Q93, D95, E97, and L99 (Figure 4F). The mutations at position L92 had the least effect, with only a slight improvement in the production of Z,Z-FOH in mutants L92A and L92F, and in the cases of L92S and L92W, the decrease in titer was observed. D95A and D95P mutants showed the same positive effects on Z,Z-FOH production. Q93 and E97 variants greatly improved the production of Z,Z-FOH. Among these mutations, the Q93S variant of Mvan4662 exhibited the highest titer of Z,Z-FOH, reaching 485.17 mg/L, increasing 550% compared to the wildtype. Our findings suggested that targeted mutagenesis can be a useful approach for optimizing the catalytic efficiency of enzymes involved in terpenoid biosynthesis. The Q93S variant of Mvan4662 may be a promising candidate for further study and potential industrial applications.

### 3.4. Batch Fermentation for Z,Z-FOH Production in E. coli

Farnesol, smiliar to other sesquiterpenoids, has been shown to be effective in inhibiting the growth of microbes such as *Candida albicans* by causing damage to bacterial cell walls and promoting cellular content release, thereby increasing bacterial susceptibility to antibiotic action [34]. To decrease the toxicity of the product and reduce volatilization, employing two-phase fermentation is a common strategy for the biosynthesis of sesquiterpenoids in *E.coli*. In our study, we chose the optimal strain ELMY101 for batch fermentation in a shake flask, and we used 10% (*v*/*v*) dodecane as the upper organic phase to capture Z,Z-FOH during fermentation.

During the first 36 h of fermentation, the glucose concentration decreased rapidly while the biomass continued to rise, reaching approximately OD_600_ of 6 and then remaining stable (Figure 5). During this period, the Z,Z-FOH production rate was relatively high. The yield of Z,Z-FOH was 0.058 g/g glucose at 96 h. By 120 h, the glucose had been depleted. The production of Z,Z-FOH also peaked and remained stable, with a maximum titer of 572.13 mg/L. To date, this is the highest titer of Z,Z-FOH synthesized by *E. coli* in a shake flask.

## 4. Discussion

We successfully constructed a de novo biosynthetic pathway of Z,Z-FOH in *E. coli*. NPPS, Z-FPPS and phosphatase are responsible for the biosynthesis of Z,Z-FOH from DMAPP and IPP through MVA pathway (Figure 1).

The Z-FPPS, Mvan4662 was previously reported to exclusively accept GPP for synthesizing Z,E-FPP in vitro [23,24]. Here, our research revealed that Mvan4662 also exhibited excellent Z,Z-FPP synthase activity when NPP, cis isomer of GPP, was used as substrate which was produced by the overexpression of NPPS in *E. coli* (Figure 2). This discovery suggests that Z-FPPS promiscuity might be dependent on available isomeric substrates to generate products with distinct configurations.

In order to improve the production of Z,Z-FOH, Mvan4662 was mutated via enzyme engineering. After docking and modeling, the non-conserved residues C41 and Y85 were first selected for site mutation. The property of amino acid chains affected the performance of enzyme activity [32]. For C41 site, when mutated to large volume of amino acids, the activities were decreased (Leu or Met) or lost (Phe, Trp, or Tyr), and while replaced with Ser which is similar in space size to Cys, the activity was increased (Figure 4). Conversely, for Y85 site, which has a bulky benzene ring structure, its replacement with the large volume of amino acids led to increased activity (Phe) and that with the small volume of amino acids (Ser, Leu or Ala) led to decreased or lost activity (Figure 4). It indicated that C41 and Y85 of Mvan4662 are not crucial but flexible sites, and C41S and Y85F were beneficial for biosynthesis of ZZ-FPP. The flexible domain is important for the binding of substrates and catalytic function.

For the second round mutations, as expectedly, among five sites in the flexible domain of the conserved region III of Mvan4662, most of these mutations increased Z,Z-FOH production, indicating that this flexible domain indeed has a significant impact on catalytic activity [33]. The larger volume of the Gln residue at position 93 compared to the mutated residues suggested that smaller amino acid residues are advantageous for Z,Z-FOH production. However, it is not necessarily true that smaller amino acid residues at this site are better. When Gln was mutated to the smallest volume residues, Glu and Ala, there was also an increase in Z,Z-FOH production (Figure 4), but it was not as high as that of the Ser mutant strain. This indicated that the hydroxyl group in the amino acid residue at this site has an important impact on catalytic activity [32].

Isoprenoid alcohols in plants can be synthesized by isoprenoid alcohol synthases, which have a poor performance in *E. coli* [30]. Diphosphatases are abundant in *E. coli* and have been employed to produce isoprenoid alcohol. NudB or AphA was used to hydrolyze IPP and DMAPP to isoprenol and prenol [35,36]. The combined expression of AphA and NudB increased the formation of isoprenol, prenol, geraniol and farnesol [37]. For the production of E,E-farnesol in *E. coli*, phosphatases PgpB and YbjG are efficient in hydrolyzing E,E-FPP. We previously reported that NudJ was the most efficient for hydrolyzing NPP and bornyl diphosphate to produce nerol and borneol, respectively [26]. Here, we showed that among 13 diphosphatases tested, NudJ was also efficient in hydrolyzing Z,Z-FPP to Z,Z-FOH (Figure 3). It was possible that phosphatase NudJ might be fitted for the production of Z-Z-isoprenoids.

In summary, we, for the first time, constructed Z,Z-FOH producing *E. coli* by rational enzyme design and metabolic engineering. We screened Mvan4662 as the most effective enzyme for producing Z,Z-FPP, which was then hydrolyzed by endogenous phosphatase in *E. coli* to form Z,Z-FOH. We identified non-conserved residues C41 and Y85 of active cavity as flexible sites. For C41S and Y85F, there was an improved production of Z,Z-FOH. Furthermore, a small library of point mutation of the flexible domain in region III showed positive efficacy on the enzyme performance, and Q93S mutant significantly improved the production of Z,Z-FOH. NudJ, an endogenous phosphatase, is a suitable enzyme for the dephosphorylation of cis-isoprenyl diphosphate and greatly increased formation of Z,Z-FOH. The engineered *E. coli* produced 572.13 mg/L Z,Z-FOH at a shake-flask level. This study provided a promising alternative for microbial cell factories producing cis-terpenoid alcohols.

## Figures and Tables

**Figure 1 microorganisms-11-01583-f001:**
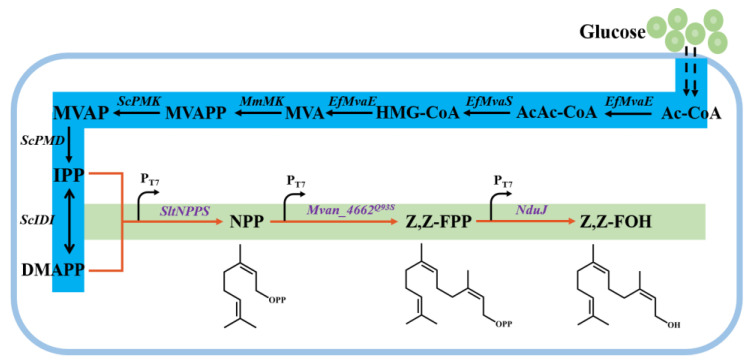
Metabolic pathway for biosynthesis of Z,Z-FOH in *E. coli*. Blue background represents biosynthesis of DMAPP and IPP in EIP81 platform strain, green background represents biosynthesis of Z,Z-FOH on base of EIP81. Notation: Ac-CoA, acetyl coenzyme A; AcAc-CoA, acetoacetyl coenzyme A; HMG-CoA, 3-hydroxy-3-methylglutaryl coenzyme A; MVA, mevalonate; MVAPP, mevalonate-5-pyrophosphate; MVAP, mevalonate-5-phosphate; IPP, isopentenyl pyrophosphate; DMAPP, dimethylallyl pyrophosphate; NPP, neryl diphosphate; Z,Z-FPP, (2Z,6Z)-farnesyl diphosphate; Z,Z-FOH, (2Z,6Z)-farnesol. Enzymes in Z,Z-FOH biosynthetic pathway: MvaE, acetoacetyl-coenzyme A thiolase; MvaS, mevalonate synthase; MK, mevalonate kinase; PMK, phosphomevalonate kinase; PMD, diphosphomevalonate decarboxylase; IDI, IPP isomerase; tNPPS, truncated NPP synthase; Ef, *Enterococcus faecalis*; Mm, *Methanosarcina mazei*; Sc, *Saccharomyces cerevisiae*; Sl, *Solanum lycopersicum*.

**Figure 2 microorganisms-11-01583-f002:**
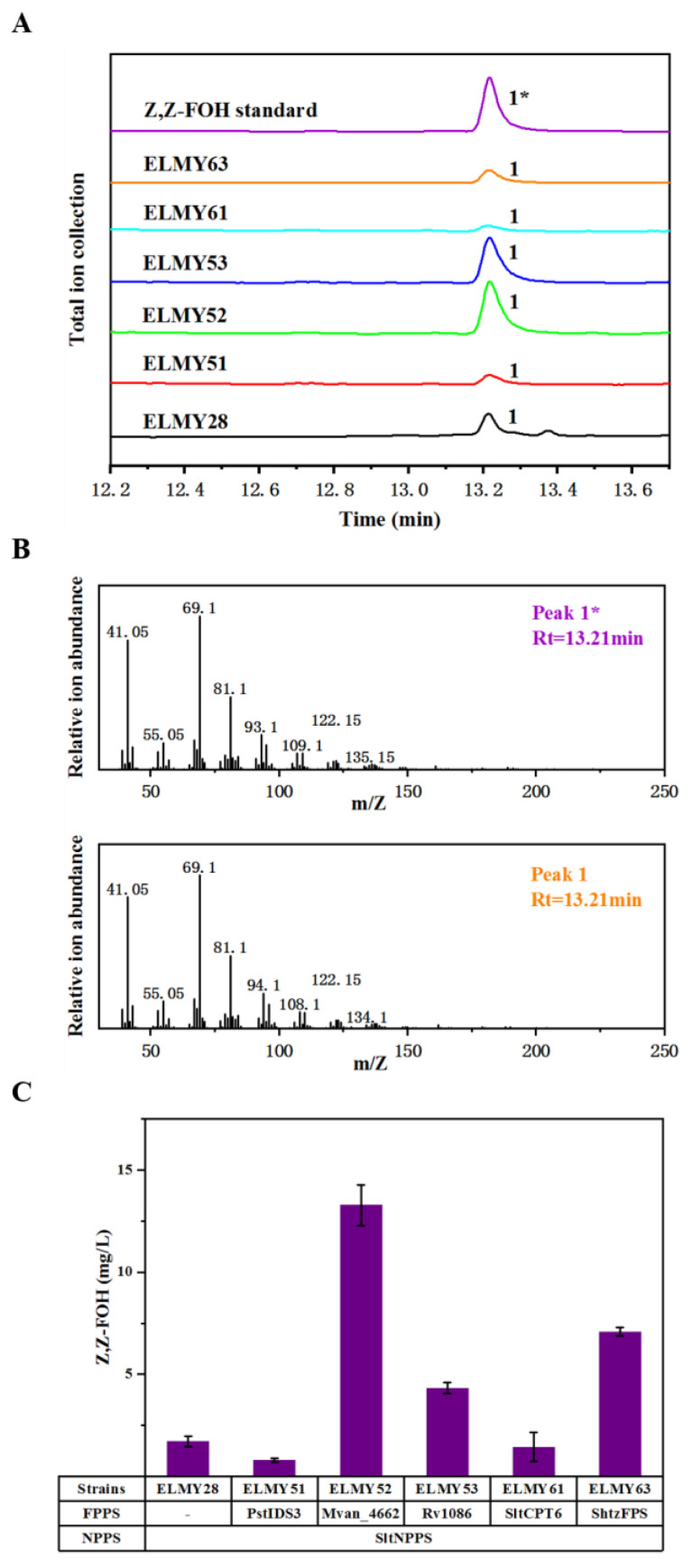
Biosynthesis of sesquiterpene Z,Z-FOH in *E. coli*. (**A**) GC chromatogram of cultures by engineered strains and Z,Z-FOH standard. (**B**) Mass spectra and retention times of peak 1* and peak 1. (**C**) Z,Z-OH production by candidate strains. Rt, retention time (min).

**Figure 3 microorganisms-11-01583-f003:**
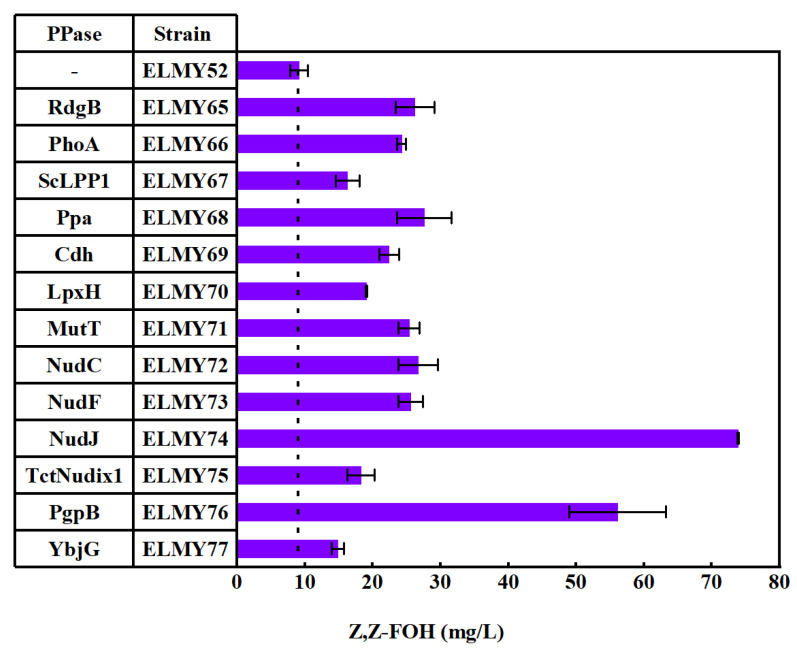
Z,Z-FOH production by employing different PPases in *E.coli*.

**Figure 4 microorganisms-11-01583-f004:**
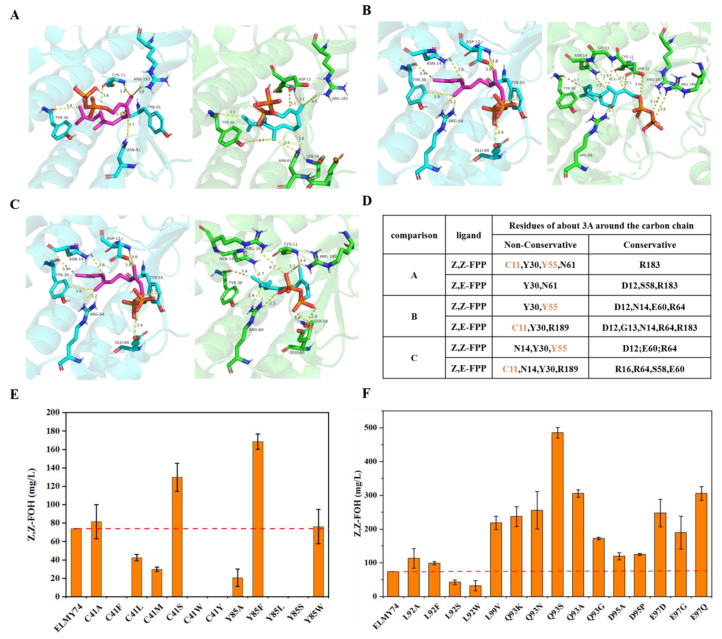
Engineered Mvan4662 for Z,Z-FOH production. (**A**–**C**) Comparisons of different docking models. Left blue backgrounds represent docking results of Z,Z-FPP and the protein template, right green backgrounds represent docking results of Z,E-FPP and the protein template. (**D**) residues that are around the carbon chain but different and non-conserved between different docking results. The first 30 amino acids have been cut in molecular docking. Orange fonts represent the candidate amino acid residues. (**E**) Z,Z-FOH production of Mvan4662 variants with altering residues that are around the carbon chain but different and non-conserved between different docking results. (**F**) Z,Z-FOH production of Mvan4662 variants with altering residues that are located in flexible domain of conserved region III.

**Figure 5 microorganisms-11-01583-f005:**
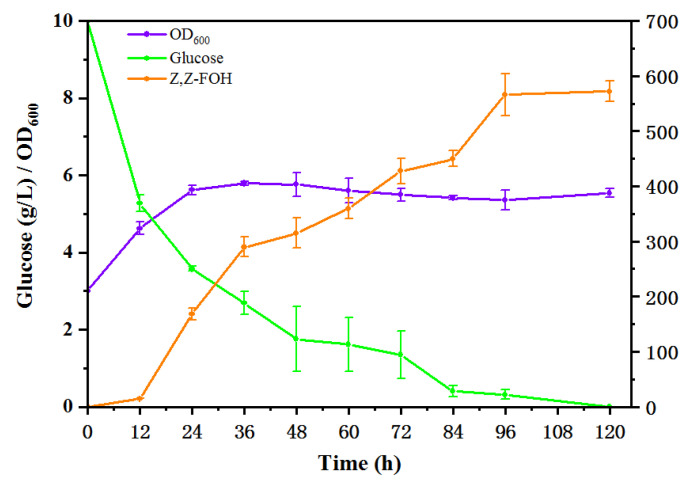
Batch fermentation processes of strain ELMY101 for Z,Z-FOH production in a shake flask.

**Table 1 microorganisms-11-01583-t001:** Strains used in this study.

Strains	Description	Source
BW25113(DE3)	*E. coli* BW25113 with T7 RNA polymerase gene in the chromosome	[26]
EIP81	BW25113(DE3) harboring plasmids pMAP6 and pISP5	[26]
ELMY28	EIP81 harboring plasmid pLMY28	This study
ELMY51	ELMY28 harboring plasmid pLMY51	This study
ELMY52	ELMY28 harboring plasmid pLMY52	This study
ELMY53	ELMY28 harboring plasmid pLMY53	This study
ELMY57	ELMY28 harboring plasmid pLMY57	This study
ELMY61	ELMY28 harboring plasmid pLMY61	This study
ELMY63	ELMY28 harboring plasmid pLMY63	This study
ELMY65	ELMY28 harboring plasmid pLMY65	This study
ELMY66	ELMY28 harboring plasmid pLMY66	This study
ELMY67	ELMY28 harboring plasmid pLMY67	This study
ELMY68	ELMY28 harboring plasmid pLMY68	This study
ELMY69	ELMY28 harboring plasmid pLMY69	This study
ELMY70	ELMY28 harboring plasmid pLMY70	This study
ELMY71	ELMY28 harboring plasmid pLMY71	This study
ELMY72	ELMY28 harboring plasmid pLMY72	This study
ELMY73	ELMY28 harboring plasmid pLMY73	This study
ELMY74	ELMY28 harboring plasmid pLMY74	This study
ELMY75	ELMY28 harboring plasmid pLMY75	This study
ELMY81	ELMY28 harboring plasmid pLMY81	This study
ELMY82	ELMY28 harboring plasmid pLMY82	This study
ELMY83	ELMY28 harboring plasmid pLMY83	This study
ELMY84	ELMY28 harboring plasmid pLMY84	This study
ELMY85	ELMY28 harboring plasmid pLMY85	This study
ELMY86	ELMY28 harboring plasmid pLMY86	This study
ELMY87	ELMY28 harboring plasmid pLMY87	This study
ELMY88	ELMY28 harboring plasmid pLMY88	This study
ELMY89	ELMY28 harboring plasmid pLMY89	This study
ELMY90	ELMY28 harboring plasmid pLMY90	This study
ELMY91	ELMY28 harboring plasmid pLMY91	This study
ELMY92	ELMY28 harboring plasmid pLMY92	This study
ELMY93	ELMY28 harboring plasmid pLMY93	This study
ELMY94	ELMY28 harboring plasmid pLMY94	This study
ELMY95	ELMY28 harboring plasmid pLMY95	This study
ELMY96	ELMY28 harboring plasmid pLMY96	This study
ELMY97	ELMY28 harboring plasmid pLMY97	This study
ELMY98	ELMY28 harboring plasmid pLMY98	This study
ELMY99	ELMY28 harboring plasmid pLMY99	This study
ELMY100	ELMY28 harboring plasmid pLMY100	This study
ELMY101	ELMY28 harboring plasmid pLMY101	This study
ELMY106	ELMY28 harboring plasmid pLMY106	This study
ELMY107	ELMY28 harboring plasmid pLMY107	This study
ELMY108	ELMY28 harboring plasmid pLMY108	This study
ELMY109	ELMY28 harboring plasmid pLMY109	This study
ELMY110	ELMY28 harboring plasmid pLMY110	This study
ELMY112	ELMY28 harboring plasmid pLMY112	This study
ELMY113	ELMY28 harboring plasmid pLMY113	This study

**Table 2 microorganisms-11-01583-t002:** Plasmids used in this study.

Plasmid	Description	Source
pMAP6	pETDuet-1, P_T7_-*EfmvaE*-P_T7_-*EfmvaS^A110G^*, Amp^R^	[8]
pISP5	pCDFDuet-1, P_T7_-*MmMK*-*ScPMK*-*ScPMD*-*ScIDI*, Str^R^	[8]
pLMY28	pACYCDuet-1, P_T7_-*SltNPPS*, Chl^R^	This study
pLMY51	pRSFDuet-1, P_T7_-*PstIDS3*, Kan^R^	This study
pLMY52	pRSFDuet-1, P_T7_-*mvan4662*, Kan^R^	This study
pLMY53	pRSFDuet-1, P_T7_-*rv1086*, Kan^R^	This study
pLMY57	pRSFDuet-1, P_T7_-*IspA-rv1086*, Kan^R^	This study
pLMY61	pRSFDuet-1, P_T7_-*SltCPT6*, Kan^R^	This study
pLMY63	pRSFDuet-1, P_T7_-*ShtZFPS*, Kan^R^	This study
pLMY65	pRSFDuet-1, P_T7_-*mvan4662*-P_T7_-*rdgB*, Kan^R^	This study
pLMY66	pRSFDuet-1, P_T7_-*mvan4662*-P_T7_-*phoA*, Kan^R^	This study
pLMY67	pRSFDuet-1, P_T7_-*mvan4662*-P_T7_-*ScLpp1*, Kan^R^	This study
pLMY68	pRSFDuet-1, P_T7_-*mvan4662*-P_T7_-*ppa*, Kan^R^	This study
pLMY69	pRSFDuet-1, P_T7_-*mvan4662*-P_T7_-*cdh*, Kan^R^	This study
pLMY70	pRSFDuet-1, P_T7_-*mvan4662*-P_T7_-*lpxH*, Kan^R^	This study
pLMY71	pRSFDuet-1, P_T7_-*mvan4662*-P_T7_-*mutT*, Kan^R^	This study
pLMY72	pRSFDuet-1, P_T7_-*mvan4662*-P_T7_-*nudC*, Kan^R^	This study
pLMY73	pRSFDuet-1, P_T7_-*mvan4662*-P_T7_-*nudF*, Kan^R^	This study
pLMY74	pRSFDuet-1, P_T7_-*mvan4662*-P_T7_-*nudJ*, Kan^R^	This study
pLMY75	pRSFDuet-1, P_T7_-*mvan4662*-P_T7_-*TctNudix1*, Kan^R^	This study
pLMY76	pRSFDuet-1, P_T7_-*mvan4662*-P_T7_-*pgpB*, Kan^R^	This study
pLMY77	pRSFDuet-1, P_T7_-*mvan4662*-P_T7_-*ybjG*, Kan^R^	This study
pLMY81	pRSFDuet-1, P_T7_-*mvan4662*^C41A^-P_T7_-*nudJ*, Kan^R^	This study
pLMY82	pRSFDuet-1, P_T7_-*mvan4662*^C41F^-P_T7_-*nudJ*, Kan^R^	This study
pLMY83	pRSFDuet-1, P_T7_-*mvan4662*^C41L^-P_T7_-*nudJ*, Kan^R^	This study
pLMY84	pRSFDuet-1, P_T7_-*mvan4662*^C41M^-P_T7_-*nudJ*, Kan^R^	This study
pLMY85	pRSFDuet-1, P_T7_-*mvan4662*^C41S^-P_T7_-*nudJ*, Kan^R^	This study
pLMY86	pRSFDuet-1, P_T7_-*mvan4662*^C41W^-P_T7_-*nudJ*, Kan^R^	This study
pLMY87	pRSFDuet-1, P_T7_-*mvan4662*^C41Y^-P_T7_-*nudJ*, Kan^R^	This study
pLMY88	pRSFDuet-1, P_T7_-*mvan4662*^D95A^-P_T7_-*nudJ*, Kan^R^	This study
pLMY89	pRSFDuet-1, P_T7_-*mvan4662*^D95P^-P_T7_-*nudJ*, Kan^R^	This study
pLMY90	pRSFDuet-1, P_T7_-*mvan4662*^D95S^-P_T7_-*nudJ*, Kan^R^	This study
pLMY91	pRSFDuet-1, P_T7_-*mvan4662*^E97D^-P_T7_-*nudJ*, Kan^R^	This study
pLMY92	pRSFDuet-1, P_T7_-*mvan4662*^E97G^-P_T7_-*nudJ*, Kan^R^	This study
pLMY93	pRSFDuet-1, P_T7_-*mvan4662*^E97Q^-P_T7_-*nudJ*, Kan^R^	This study
pLMY94	pRSFDuet-1, P_T7_-*mvan4662*^L92A^-P_T7_-*nudJ*, Kan^R^	This study
pLMY95	pRSFDuet-1, P_T7_-*mvan4662*^L92F^-P_T7_-*nudJ*, Kan^R^	This study
pLMY96	pRSFDuet-1, P_T7_-*mvan4662*^L92S^-P_T7_-*nudJ*, Kan^R^	This study
pLMY97	pRSFDuet-1, P_T7_-*mvan4662*^L92W^-P_T7_-*nudJ*, Kan^R^	This study
pLMY98	pRSFDuet-1, P_T7_-*mvan4662*^L99V^-P_T7_-*nudJ*, Kan^R^	This study
pLMY99	pRSFDuet-1, P_T7_-*mvan4662*^Q93K^-P_T7_-*nudJ*, Kan^R^	This study
pLMY100	pRSFDuet-1, P_T7_-*mvan4662*^Q93N^-P_T7_-*nudJ*, Kan^R^	This study
pLMY101	pRSFDuet-1, P_T7_-*mvan4662*^Q93S^-P_T7_-*nudJ*, Kan^R^	This study
pLMY106	pRSFDuet-1, P_T7_-*mvan4662*^Y85A^-P_T7_-*nudJ*, Kan^R^	This study
pLMY107	pRSFDuet-1, P_T7_-*mvan4662*^Y85F^-P_T7_-*nudJ*, Kan^R^	This study
pLMY108	pRSFDuet-1, P_T7_-*mvan4662*^Y85L^-P_T7_-*nudJ*, Kan^R^	This study
pLMY109	pRSFDuet-1, P_T7_-*mvan4662*^Y85S^-P_T7_-*nudJ*, Kan^R^	This study
pLMY110	pRSFDuet-1, P_T7_-*mvan4662*^Y85W^-P_T7_-*nudJ*, Kan^R^	This study
pLMY112	pRSFDuet-1, P_T7_-*mvan4662*^Q93A^-P_T7_-*nudJ*, Kan^R^	This study
pLMY113	pRSFDuet-1, P_T7_-*mvan4662*^Q93G^-P_T7_-*nudJ*, Kan^R^	This study

## Data Availability

All data are present in the manuscript and its Appendix A.

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
