# Peer review of "Metabolic Engineering for Efficient Production of Z,Z-Farnesol in E. coli"

_microorganisms, 2023, doi:10.3390/microorganisms11061583_

Round 1
Reviewer 1 Report
The study is well structured and for the most part of it also described clearly and to sufficient extent. I have some incidental as well as general comments that should be addressed before this manuscript can be accepted:
line 31: I think it should be high [energy] density. Also, that sentence could use a couple of references.
line 35: should read "resource intense and therefore inefficient and hence expensive" I believe.
line 36: "heavy pollution" could be differentiated a bit better and also backed with some meaningful references.
line 42: should read "fermentable carbon sources", I believe.
line 69: I'm unsure "suppled" is actually a word - do you mean "supplied" or maybe "supplemented"?
line 71: I'm quite sure Mycobacterium tuberculosis is, in fact, a pathogen. I'd just remove the word "nonpathogenic"; it doesn't really matter in this context.
line 80: "various" is rather unspecific - do you mean "rational engineering strategies" or maybe "different engineering approached" or "a modular engineering approach"?
line 105: reconstruction sounds like you did it before, which isn't true I believe - just say construction.
line 111: ... and are listed...
line 112-119: use italic-font for species names
line 226: hydrolysis of
line 262: strictly conserved
line 267: flexibility
line 293: conserved
line 332: unconserved
line 334: anything measured in g/L is a titer, not a yield. Make sure to differentiate these process metrics clearly and strictly. Furthermore, I would be interested in the yields (per substrate) too, if those were determined. That somewhat relates to another one of my concerns: I am wondering why the M9 medium contained 2 g/L yeast extract and how that impacted cultivation / production (titer/yield)? Were the appropriate controls conducted to rule out that the product was derived from the complex substrates rather than glucose?
Further, the discussion is overall rudimentary and does not suffice - it reads more like a conclusion. A proper discussion should explain the obtained results in light of existing literature. Some of that is present in the results section (e.g., the explanation as to why certain mutations are beneficial, while others completely deactivate Mvan4662). I suggest separating results and discussion better and also focus some more on explaining the observations and implications thereof for future work.
I also noted that the strains in figure 2 are named BLMY, while in most (all?) other places strains are named ELMY - are these the same strains?
Sometimes the grammar is a bit off, also sometimes wrong words are used.
Reviewer 2 Report
The study focuses on metabolic engineering of E. coli for production of Z, Z-Farnesol. The authors have introduced a de novo pathway for its production and optimized the production by combinatorial and rational engineering approaches. The uniqueness of study is that it is a proof-of-concept study and Z, Z-Farnesol production is reported for the first time in E. coli, therefore paving the way towards microbial based production of this metabolite. Here are a few comments that authors must address before publication:
- What is the effect of using four antibiotics on growth? Now the authors have winner enzyme candidates it will make more sense to integrate these shortlisted genetic parts into genome and checking ZZ-Farnesol production in absence of antibiotics. The industrial processes demand plasmid free and antibiotic free strains for stability and cost-effectiveness.
- What is the reason behind using truncated version of enzymes instead of full enzymes?
- Figure 2: What is the control in this experiment? Did the authors check if this peak was present in ELMY28? Also, peak in ELMY61 looks sharper in Fig 2a but the quantification of ZZ-Farnesol looks quite low compared to ELMY52.
- Fig 2a and Fig 2c has different name for strains.
- Figure 2: The enzymes screened were specific for producing ZZ-FPP and endogenous phosphates are converting the intermediate ZZ-FPP to ZZ-FOH. Did the author also quantified ZZ-FPP to get an estimate of intermediate product conversion. The quantification seems important considering it will provide information How much efficient is the new phosphate and how much scope is still present to further improve the process by converting the entire ZZ-FPP to ZZ-FOH.
- Is it a first report of Z, Z-Farnesol production in E. coli or across all other prokaryotic host? It is not clear from introduction.
- Explain two phase fermentation. I specifically missed How it was performed and what authors mean by this approach.
- Line 30-32: Provide reference
- Line 80: Instead of writing “various engineering strategy” mention more specific terms like metabolic engineering, rational engineering, targeted engineering etc.
- Line 160: The authors mentioned use of chloramphenicol as one of antibiotic selection. But it is not clear from strain table which strain has Chloramphenicol resistance.
- Line 164: Starting OD of 3 seems very high considering it is E. coli and it is in minimal medium.
- Line 172: Metabolite was quantified seems quite general. Is it just metabolite (Z, Z-Farnesol) or multiple metabolites were analyzed?
- Line 237: The use of word “fortunately” seems inappropriate. Interestingly seems more appropriate.
- Line 374: The sentence “Our breakthrough technology” seems inappropriate since it is not a novel technology, and it is simply use of metabolic engineering approaches which is widely employed to improve product titers in microbes.
Round 2
Reviewer 2 Report
The authors have provided point to point response for all my comments to my satisfaction. I have no further comments for authors.